# Intended and experienced literacy practices in a Swedish undergraduate nursing education

**Maria Christidis**[ID][1,2*], **Nikolaos Christidis**[ID][3], **Viveca Lindberg**[4], **Kristina Gottberg**[ID][2], **Carina Georg**[2]

1 Department of Nursing Science, Sophiahemmet University, Stockholm, Sweden, 2 Department of Neurobiology, Care Sciences and Society, Karolinska Institutet, Huddinge, Sweden, 3 Division of Oral Rehabilitation, Department of Dental Medicine, Karolinska Institutet, Huddinge, Sweden, 4 Department for Teaching and Learning, Stockholm University, Stockholm, Sweden

* maria.christidis@ki.se

## Abstract

### Introduction

Academic literacy in higher education has been widely studied, but less attention has been given to literacy practices within professional programmes such as nursing education. This study aimed to analyze one Swedish undergraduate nursing programme regarding the presence of academic and professional literacy, and secondarily to explore students' note-taking as a component of literacy practices.

### Methods

The study employed a descriptive and exploratory design, analyzing curriculum documents and surveying nursing students. An analysis of the intended curriculum (course syllabi) of a three-year undergraduate nursing programme at Karolinska Institutet was conducted to identify explicit and implicit literacy components. A digital questionnaire focusing on note-taking practices was distributed to second-semester students (n = 67; response rate 40%). Closed questions were analyzed using descriptive statistics, while open-ended responses underwent qualitative content analysis.

### Results

The curriculum analysis demonstrated that academic literacy was primarily addressed through scientific writing, group projects, and thesis work, particularly in the first two and final two semesters. Professional literacy was integrated across all semesters and included communication with patients, families, and interprofessional teams, documentation, and interpretation of professional texts. Surveyed students reported frequent note-taking, primarily during lectures and prior to examinations, using both pen and paper and digital devices, with a preference for pen and paper. Qualitative data indicated that students use note-taking mainly to support

**Data availability statement:** All relevant data are within the paper and its Supporting Information files.

**Funding:** The author(s) received no specific funding for this work.

**Competing interests:** The authors have declared that no competing interests exist.

**Abbreviations:** REDCap, Research Electronic Data Capture; RN, Registered Nurse; SFS, Svensk FörfattningsSamling (Swedish Code of Statutes); SOLO, Structure of the Observed Learning Outcome; SPN, Study Programme in Nursing.

memorization, understanding, and exam preparation, with limited focus on professional literacy needs.

## Conclusion

The nursing curriculum integrates both academic and professional literacy practices, although with different emphases across the study period. Students' current note-taking practices are predominantly academically oriented. These findings highlight the need for more explicit integration and scaffolding of both academic and professional literacy throughout nursing education to better prepare students for their future professional roles, i.e., for clinical communication, documentation, and interprofessional collaboration essential to safe and effective patient care.

## Introduction

Academic literacy has been widely discussed in higher education, but less often in connection to professional education and the literacy required in the future professions. Here. academic literacy refers to differences in *what* people read and write, *how* they do it and for *what purposes* (why). These practices vary across social contexts, including disciplines and related programmes [1–3]. This perspective highlights the diversity of literacy practices that students encounter during key transitions in their educational journey, such as from upper secondary school to academic higher educations. Students transitions have been studied, with focus on identity formation and the role of context and authenticity in shaping students' understanding of relevance [2]. Furthermore, the issue of different literacy practices and students' initial problems with coping with 'academic' writing concerning expectations on reading and writing has been addressed previously [4,5]. Concerning nurse education, specific expressions of the need for professional literacy was for instance found in the Swedish Government Official Reports in terms of documentation of nursing interventions and patient status from 1948 (SOU 1948:17) [6]. This is, however, an area that has been further developed since then, and goes beyond the example given of documentation.

This approach to literacy relates to reading and writing as aspects of social practices, thus indicating that students entering higher professional education may be well acquainted with literacy practices that they have previous experiences of, but they are still newcomers to the kind of literacy practices demanded within a specific educational programme. This view has been further elaborated on in that it becomes more meaningful and relevant for the individual when literacy is related to a specific context and to a future profession [7]. In this way, literacy becomes authentic and relevant for diverse students aspiring to become professionals in different fields. When it comes to studies on academic literacies within higher education, this has been explored in a longitudinal comparison between subjects history and economics in Sweden [8], while internationally, theoretical and methodological contributions have been made [1,9–11].

In relation to nursing education, there is a need to call for academic literacy studies [12], which is further confirmed by a recent systematic review [13]. However, since professional higher education prepares for specific professions, the aspect of professional literacy practices is equally important to attend to in nursing education. Professional literacy involves questions such as: what kinds of texts are typical and recurring within a specific profession? What are characteristic features (content and form) of these texts? What are the purposes of these texts? The answers to these questions differ between professions (on engineers' professional writing [14]). Examples of studies that focus on professional higher education represent engineering education [14,15], teacher education [4], police education [16], dental education [17,18]. A few studies have focused on both literacy practices for academia and the future profession [19]. Even fewer have simultaneously studied literacy practices that students are expected to appropriate for both a specific educational programme and their future profession [14,20–22].

An early example of this kind of research looked into nursing and engineering [23], where literacy within nursing and engineering education was studied and compared to the literacy practices within the respective professions. This study indicated that there were substantial differences between the literacy practices in education and workplaces. This raises the question of how higher education prepares students for the double literacy task, that is, academic literacy for academic achievement, and professional literacy for the future profession [14,20]. Studies that specifically address professional literacy within nursing are also few. Documentation as a professional literacy practice has been acknowledged previously, in relation to the need of improving this also for experienced nurses [24]. Concerning the medical professions, research emphasizes the importance of accurate and complete medical records in healthcare. Variations in record formats depend on differences between countries, professional roles, and individual experiences [25]. Deficiencies in record keeping can negatively affect patient safety. Studies show that electronic patient records (EPR) and standardized templates can improve the quality of documentation [26,27]. Auditing and training have also been shown to improve surgical notes and increase compliance with guidelines [28]. Of course, academic literacy may also be addressed as a competence for future work, since today nurses are expected to be able to follow the scientific advancement of their professional knowledge-base, adopting a critical approach [29]. Furthermore, a number of nurses will eventually apply for postgraduate educations and become researchers.

Nevertheless, the literacy practices needed for professional education and practice are not only an issue for the individual student, instead it is a reciprocal issue for the educational programme *and* the student: how are the literacy practices made visible in the syllabi for the courses in the undergraduate nursing education? How do students describe what, how and for what purposes they write during their studies? While a predominant picture of international research within the field of New Literacy is based on textual data like interview transcripts, samples of student-produced texts or teachers' feedback on these texts as well as in Swedish studies [4,8,10,14], this study takes a different point of departure. In line with a previous study who analyzed the undergraduate dental education for indications of literacy demands (the intended curriculum [30]) related to academia as well as dentistry [21], this study will complement the research field both empirically and methodologically. Thus, the aim of this study was to analyze one of the Swedish undergraduate nursing programmes, with respect to possible content of relevance for academic and professional literacy. Secondarily, to describe note-taking as an aspect of literacy practices, from the perspective of nursing students.

## Materials and methods

### Study context

The study design was both descriptive and exploratory [31], as it included an analysis of the nursing programme's course syllabi and a questionnaire addressing aspects of academic and professional literacy in teaching at the nursing programme, completed by nursing students.

This study was approved by the Swedish Ethical Review Authority with (Dnr. 2022-04816-01), and conducted at Karolinska Institutet in Sweden, at the study programme in nursing (SPN), coded as 1SJ18 (Programme syllabus

- Sjuksköterskeprogrammet, 180 hp | Karolinska Institutet Utbildning). In Sweden the SPN is a three-year post-secondary education, comprising 180 ECTS, that qualifies for a license to practice as a registered nurse (RN). The SPN awards a Degree of Bachelor of Science in Nursing and a Degree of Bachelor of Medical Science with a major in nursing. The SPN curriculum integrates theoretical courses as well as clinical courses. This allows students to develop knowing-in-action while observing how nurses interact with patients, colleagues, and both linguistic and material resources. A considerable share of the program is dedicated to clinical education that takes place in various health care settings (SFS 1993:100). Learning in nursing education is complex. Undergraduate students must develop both clinical and academic skills, as well as broader professional competences [32].

## Object of analysis

The educational plan of the SPN was analyzed according to the two following analytical questions and structured in a table (Table 1). The analytical questions were:

1. What are the explicit and implicit expressions of writing practices in the SPN?

2. How do the expected learning outcomes and content of the course syllabi align with expressions of academic and professional writing practices in the SPN?

## The questionnaire regarding students' literacy practices

A questionnaire was developed to map students' note-taking as an aspect of literacy practices during their education. For the development of this we used results from previous inter/national research [18,33,34] for comparative purposes. It included closed questions on how, when and for which purpose students take notes and what tools students use for note-taking (n = 10). The answering format was on an ordinal level with the alternatives, i.e., "always, sometimes, seldom or never". In addition, open-ended questions (n = 9) with the possibility to comment or explain literacy practices were included. The questionnaire was distributed via Research Electronic Data Capture (REDCap) web application [35,36] to nursing students in their second semester of the SPN during the autumn of 2023, starting 2023-10-01 and ending 2023-12-31 after three reminders (S1 File – Appendix 1a and S2 File – Appendix 1b). The students were informed in written form about their right to withdraw at any time without the need for giving a reason and that their questionnaire answers had no connection with grades or study performance within the SPN [37]. The students were informed about the study through the learning management system and invited to participate, and informed consent was obtained by means of answering the digital questionnaire.

## Analytical methods

Descriptive statistics were used to present quantitative data from the digital questionnaire. For closed questions frequency distribution was used, complemented with mean and range for age [38], while for the open-ended questions, a qualitative content analysis [39] was applied to analyze and describe the results on literacy practices. First, all the written answers with free text (displayed in S3 File – Appendix 1c) were collected per question and reviewed several times. Secondly, for each question, the transcripts were grouped according to the content into subcategories. Thirdly, the answers to open-ended questions were reviewed in relation to answers provided by the quantitative questions for any clarifications regarding differences in practices between, for instance, purposes for note-taking. In the last step of analysis, the preliminary results were reviewed and discussed within the research team and categories were determined.

## Statements

**Ethics approval and consent to participate:** This study was approved by the Swedish Ethical Review Authority with (Dnr. 2022-04816-01).

**Table 1. Courses in the Study Programme in Nursing, at Karolinska Institutet, with learning objectives related to academic or professional literacy.**

| Semester | Focus on professional literacy | Focus on academic literacy |
|---|---|---|
| **1ˢᵗ semester** | | |
| ***The Nurse Profession and Nursing Science*** *16.5 credits* | • Communication for good nursing care<br>• Engage in ethical reasoning in relation to different healthcare situations | • Apply scientific writing<br>• Apply reference management according to APA |
| ***Anatomy and Physiology*** *13.5 credits* | • Be able to use anatomical terminology | |
| **2ⁿᵈ semester** | | |
| ***Research Methods*** *6 credits* | | • Plan, implement, present and evaluate project work in group collaboration<br>• Apply scientific language, accepted writing rules and reference management |
| ***Foundation of Nursing Care in Health and Illness*** *24 credits* | • Describe different communication models<br>• Distinguish information to be recorded regarding the patient's care<br>• Communicate in speech and writing in a patient-safe way and use concepts and expressions commonly used in health care<br>• Demonstrate oral and written communication skills based on the nursing profession<br>• Demonstrate the ability, through communication and an empathetic approach, to create a nursing relationship that considers the patient's and relatives' need for participation | • Reflect on and compile scientific studies for evidence-based nursing in health-promoting care |
| **3ʳᵈ semester** | | |
| ***Nursing Care in Illness and Disease*** *30 credits* | • Describe the principles of secure patient information transfer and explain the laws, regulations and record keeping<br>• Communicate in speech and writing in a patient-safe way and use concepts and expressions commonly used in health care<br>• Assess, analyze, implement, evaluate, and record nursing care<br>• Report orally and in writing information regarding the patient's care<br>• Be able to provide individualized and situation-specific information to patients and relatives that promotes learning and participation based on individual needs<br>• Implement teaching that supports patient, family, and colleague learning<br>• Communicate and collaborate in interprofessional teams<br>• Perform calculations so that medicines are dosed and diluted correctly | |
| **4ᵗʰ semester** | | |
| ***Nursing Care in Mental Health and Psychiatry*** *12 credits* | • Communicate in speech and writing in a patient-safe way and use concepts and expressions commonly used in health care<br>• Care for, respond to and communicate with patients based on the principle of health and care on equal terms<br>• Assess, plan, implement, document, and evaluate individual nursing care for mental illness and disease based on the person's needs, resources and health obstacles<br>• Communicate with and collaborate within their own profession and with other professions in the planning and implementation of care and nursing in cases of mental illness | |
| ***Nursing Care of Children and Adults in Primary Care and Home Health Care*** *13.5 credits* | • Understand and reflect on communication and assessment of the individual's nursing needs in primary care<br>• Understand and reflect on communication with children and parents in relation to nursing interventions<br>• Communicate in speech and writing in a patient-safe way and use concepts and expressions commonly used in health care<br>• Apply knowledge of communication in the encounter with patients of all ages and their relatives in primary care, home care and child health care/health care contexts<br>• Apply pedagogical skills to support patient learning<br>• Communicate and cooperate in interprofessional teams<br>• Take and give constructive criticism and identify self-development needs | |

*(Continued)*

**Table 1.** (Continued)

| Semester | Focus on professional literacy | Focus on academic literacy |
|---|---|---|
| **5<sup>th</sup> semester** | | |
| *Critical Care Nursing*<br>*16.5 credits* | • Communicate in speech and writing in a patient-safe way and use concepts and expressions commonly used in health care<br>• Apply structured patient assessment and communication, clinical reasoning and readiness for action<br>• Apply accurate, structured and patient-safe information transfer in writing and speech<br>• Correctly calculate, prepare, administer and evaluate the efficacy of medicines, and comply with applicable laws and regulations<br>• Perform drug calculations related to acute health conditions in a patient-safe manner | |
| *Research Methods and Nursing Research*<br>*6 credits* | | • Formulate a purpose for a scientific study in a nursing-relevant area<br>• Apply accepted health care concepts and expressions as well as scientific language, accepted writing rules and reference management |
| **6<sup>th</sup> semester** | | |
| *Nursing Care of the Elderly in A Changing Life Situation*<br>*12 credits* | • Communicate in speech and writing in a patient-safe way and use concepts and expressions commonly used in health care<br>• Independently, based on evidence and a person-centered approach, lead, plan, prioritize, implement, record and evaluate nursing care for complex health problems in collaboration with the elderly and their relatives<br>• Independently, based on laws and regulations, administer and evaluate drug prescriptions and report and record drug effects and side effects<br>• Communicate in speech and writing in a patient-safe manner and use concepts and expressions that are accepted in health care | |
| *Interprofessional and Professional Competence*<br>*3 credits* | • Reflect on and argue for their own and other professions' competence for increased patient safety, and demonstrate the ability to communicate and collaborate with patients, relatives and other professions<br>• Communicate in speech and writing in a patient-safe way and use concepts and expressions commonly used in health care | |
| *Degree Project in Nursing*<br>*15 credits* | | • Present the degree project in writing with scientific accuracy and scientific language<br>• Present, discuss and defend the chosen study design, implementation, results, and conclusions of the scientific work |

**Clinical trial number:** Not applicable since it is not a clinical trial.

**Consent for participation and for publication:** The students were informed about the study through the learning management system and invited to participate, and informed consent was obtained by means of answering the digital questionnaire.

## Results

### Literacy in the curriculum for the study programme in nursing

The SPN includes courses that specifically address academic or professional literacy in their objectives, based on the idea of a taxonomy, although not specified. There are two dominating taxonomies used in the Western world, the Bloom's taxonomy for the cognitive domain [40,41], which has also been further developed [42], and the SOLO taxonomy (Structure of the Observed Learning Outcome) [43,44]. These taxonomies are based on different theoretical points of departure. The learning outcomes of these courses are presented in Table 1.

### Professional literacy

Based on the analysis of the included courses in the SPN the intended curriculum include professional literacy skills such as: 1) various communication models; 2) oral and written communication with patients in different ages; 3) oral and written communication with patients with different needs; 4) oral and written communication with relatives to patients; 5) oral and written communication with other medical professions; 6) oral and written communication with interprofessional teams; 7) writing patient records; 8) communicate patient care in speech and writing in a patient-safe way; 9) writing assessments, treatment plans, treatment progress and follow-ups and communicate this to interprofessional teams; 10) reading and interpreting patient documentation (records, plans, referrals etc.); 11) calculate, prepare, administer, and evaluate the efficacy of medicines; 12) reading and interpreting the professional meaning of laws and regulations.

For oral communication, students are expected to use the professional language taught throughout the programme. Additionally, they should demonstrate the ability to adapt to their communication according to the needs of different patients, professional collaborators, and the broader society. Furthermore, students are expected to care for, respond to, and communicate with patients of different ages, with different illnesses, and with different needs based on the principle of health and care on equal terms, ensuring respectful, empathetic, and patient-centered interactions. These skills are trained on throughout the SPN, i.e., every semester, with variations depending on patient age, illnesses, care-situation, etc., i.e., the context, both on campus and in clinical placements.

### Academic literacy

Based on the analysis of the included courses in the SPN the intended curriculum include academic literacy skills such as: 1) writing scientific texts including adequate scientific concepts and reference management; 2) planning, writing, presenting and evaluating group projects; 3) formulating a research question in a nursing specific area; 4) writing a degree project in nursing; 5) orally presenting and defending a degree project in nursing.

In contrast to professional literacy that is taught and trained every semester, the academic literacy was visible in the SPN the first two and the last two semesters, mainly with the purpose to be able to plan, write, present a scientific project and evaluate and reflect on scientific publications with an increase in depth the last two semesters where the students are writing their degree project.

### Students note-taking

In total, the questionnaire was sent to 166 students, of whom 67 responded (40% response rate). The questionnaire on students' note-taking practices is presented in Table 2. A majority of the students reported that they always (43%) or sometimes (36%) were taking notes at lectures. During these lectures, the students reported that they always (53%) and/or sometimes (26%) used a pencil, while they always (35%) and/or sometimes (35%) used a computer/tablet. In contrast, students never (59%) and/or seldom (20%) used mobile phones. After lectures students reported that they used a pencil at about a similar frequency (56%) as during lectures, while they used a computer/tablet at a higher frequency (46%) than during lectures. Also, students used mobile phones at about the same frequency (2%). Frequency regarding when students are always and/or sometimes engaged in note-taking was highest during lectures (79%) and before examinations (78%), while the frequency was lower after lectures (49%). The most used tool for note-taking in relation to lectures was pencil, while before examinations pencil and computer were equally used. The highest proportion of students taking notes after lectures were always doing this for themselves (88%), and sometimes for their study group (18%) or friend(s) (15%).

The students described that the reasons for writing these summaries were to enable their own understanding, to clarify concepts and terms, to memorize and to facilitate their preparation for examinations. From the qualitative content analysis on the students' described reasons for taking notes, two subcategories emerged within an overarching category of *Taking notes for facilitating learning.* One subcategory was taking notes for the purpose of *Summarizing to memorize and understand*, as an example that one student wrote:

**Table 2. Note-taking practices among 67 undergraduate nursing students, in the Study Programme in Nursing, at Karolinska Institutet, at different learning occasions and tools applied.**

| Age, years | | | |
|---|---|---|---|
| Mean (SD) | 33 (10) | | |
| Min – Max | 19–53 | | |
| **Educational form (%)** | | | |
| Campus-based education | 67 | | |
| Distance education | 33 | | |
| **Note-taking at lectures (%)** | | | |
| Always | 43 | | |
| Sometimes | 36 | | |
| Seldom | 9 | | |
| Never | 12 | | |
| **Tools for note-taking during lectures (%)** | | | |
| | *Pencil* | *Computer/tablet* | *Mobile* |
| Always | 53 | 35 | 2 |
| Sometimes | 26 | 35 | 19 |
| Seldom | 9 | 22 | 20 |
| Never | 12 | 7 | 59 |
| **Note-taking after lectures (%)** | | | |
| Always | 18 | | |
| Sometimes | 31 | | |
| Seldom | 16 | | |
| Never | 34 | | |
| **Tools for note-taking after lectures (%)** | | | |
| | *Pencil* | *Computer/tablet* | *Mobile* |
| Always | 56 | 46 | 3 |
| Sometimes | 26 | 32 | 18 |
| Seldom | 5 | 12 | 18 |
| Never | 14 | 10 | 61 |
| **How note-taking is performed after lectures? (%)** | | | |
| | *Individually* | *My group* | *My friends* |
| Always | 88 | 2 | 2 |
| Sometimes | 12 | 18 | 15 |
| Seldom | 0 | 22 | 30 |
| Never | 0 | 58 | 53 |
| **Note-taking summary before examinations (%)** | | | |
| Always | 33 | | |
| Sometimes | 35 | | |
| Seldom | 6 | | |
| Never | 26 | | |
| **Tools for note-taking before examinations (%)** | | | |
| | *Pencil* | *Computer/tablet* | *Mobile* |
| Always | 45 | 45 | 2 |
| Sometimes | 27 | 45 | 20 |
| Seldom | 12 | 4 | 9 |
| Never | 16 | 6 | 69 |

"Ett sätt att lära och repetera. Många gånger skriver jag anteckningar som jag sedan inte behöver läsa igen, jag lär mig genom att skriva."

"A way to learn and repeat. Often, I take notes that I do not even need to read again; I learn simply by writing." [authors' translation]

The other subcategory emerging from the content analysis was that the students took notes for the purpose of *Preparing for the written examination*. An example for this subcategory was:

"Kunna få så bred kunskap som möjligt samt se det som ett inlärningstillfälle inför examinationer."

"To gain as broad a knowledge base as possible and to view it as a learning opportunity in preparation for examinations." [authors' translation]

## Discussion

Based on the results, the following issues will be discussed: 1) the integration of literacy in professional education 2) Students' reported forms of note-taking, 3) strengths and limitations of the results, and 4) the need for further studies.

### The integration of literacy in professional education

Based on the syllabi, undergraduate nursing education is based on mainly two academic disciplines: nursing and medicine. These two disciplines differ in literacy practices. Nursing courses require reasoning with concepts, diverse epistemological perspectives, and contextual judgement. While the content courses in nursing are related to human encounters (patients, clients, colleagues) and humanistic values, ethics, peoples' experiences, demanding critical reflection, discussions and the ability to alter perspectives, medical courses are mainly related to content such as anatomy, physiology, pathology and medical calculations, basically demanding knowledge of facts but also judgement specifically related to the nursing profession. A challenge for students is that although the same verbs can be used in learning outcomes and may signal similar types of answers, the differences in expectations are specifically related to what counts as valid answers in nursing sciences as a humanistic discipline versus medical sciences as part of natural science tradition [45]. Whether these differences are made explicit to students or not cannot, however, be answered by merely analyzing documents, other empirical data are needed. The validity of texts is not determined by individual teachers but by the academic discipline itself, with its established traditions for what and how to read and write [2,46,47]. Thus, the students are expected to appropriate a discipline-related practice, and here in the context of nursing [1].

Students reported using note-taking mainly to memorize information and prepare for exams, suggesting that their practice is strongly shaped by teaching and assessment. Thus, note-taking is most plausibly related to an organized activity by the university, that is, teaching and assessment. Second-semester students showed no signs of connecting note-taking to professional needs, focusing instead on academic learning and exam preparation. Instead, there is a clear focus on the present situation, that is, the academic context in terms of learning and passing exams. This may also be a signal that students, early on the SPN, have not yet become aware of the demands of professional literacy. Indeed, the students' experience in the nursing context comprised a week of field studies of observing nursing care, which may not provide adequate preconditions for understanding the aspect of literacy. This has also been highlighted in a previous study where there was a discrepancy between students' preparedness for academic and professional literacy practices at the time for graduation and five years into their professional careers [22].

According to previous research there are dual goals within professional education, where teaching integrates both academic literacy and professional literacy, that in principle may equip students for both academic and professional success [21,24]. Also, it has been shown that academic literacy promotes critical thinking which is essential for the nursing profession [13]. In the course syllabi of this SPN, academic literacy is in the form of scientific writing and as preparation for the Bachelors' thesis. While previous studies have acknowledged the crucial role of teaching in helping students recognize this connection, the kind of professional programmes that involve knowledge from two or more different disciplines require

further investigation [8]. This SPN features similar vocabulary regarding expected learning outcomes, due to taxonomical formulations, but differ in their criteria for what counts as valid written answers [48]. In a previous study, nursing students' academic writing included learning and using an academic writing style, which required time and effort to learn [49]. When two academic literacy practices (nursing, medicine) are involved, it can be surmised that this requires attention from teachers in order for students not to experience these as differences in individual teachers' demands. Since students must also adopt a third literacy practice, the professional one, they cannot be expected to manage all three without guidance. According to a study that investigated college students' note-taking, most students report that their note-taking is self-regulated. Those who report teacher guidance, refer to middle school or high school [50]. Whether this support is offered to them or not is, however, an empirical question.

## Students' reported forms of note-taking

This study showed two forms of note-taking, pen and paper, and digital forms of note-taking, such as computers, tablets and mobile phones. A surprising result was that pen and paper was the dominating form of note-taking reported by the students, followed by note-taking on a computer/tablet.

Although digital tools for note-taking are widely available, studies show that they have not significantly changed the amount nor the quality of students' notes [50]. It is the student that makes the choice of how and what to write. Also, in line with the practice of the academic context, this study showed that the frequency of note-taking was higher during teaching and before examinations [50].

As a further illustration that the academic context plays a crucial role in the practice of note-taking, a previous study surveyed nursing students about their chosen method of note-taking during a semester [51]. The majority of students (71.2% of 217 participants) reported using electronic note-taking, while the remaining students (28.8%) used handwritten note-taking. A common reason for choosing electronic note-taking was speed, that is, the ability to keep up with the lecturer and the pace of teaching. However, students also reported downsides, such as distractions from social media and web surfing. This indicates that note-taking is strongly tied to the academic context, although individual factors may also influence students' choice of tool. However, that is another empirical question.

## Strengths and limitations

This study was conducted by a research team that represented three disciplines, and that has collaborated in the "preparation, organization, and reporting of results" [39]. This has demanded reciprocal exploration of how the concepts and theories used are understood, what counts as valid ways of analyzing and describing previous research as well as results. This concerns specifically the analysis of the course's syllabi.

The response rate of the questionnaire was relatively small (40%) despite reminders. The small sample (n < 100) allows us only to regard the quantitative result as indications. However, the size of 20–50% of responders is a common response rate of, e.g., course evaluations and other non-mandatory and anonymous evaluations at our university. There are two possible risks: 1) that the students interpret the questionnaire as yet another evaluation within the SPN, although they were informed of the purpose; and 2) that the content and purposes of the questionnaire are of little interest to the students although of great interest to the teachers. Although we followed recommendations for increasing the response rate, such as informing of the purpose in advance, using a mixture of open ended and closed questions and directing the questions to a specific group [52,53] this was obviously not enough for motivating the students. Thus, the results must be interpreted with caution. Another contributing factor to the response rate might also have been that the questions in the questionnaire were limited to note-taking in relation to lectures and examinations. This can be seen as a weakness of the study. Furthermore, the choice of the group of students and the course during the period of data collection was mostly based on teacher-led activities. Furthermore, in this article, the clinical context was not included.

**Further studies**

This article is the first one in a series of longitudinal study concerning literacy in the nursing programme. Future studies should include larger samples, follow students longitudinally, and address literacy practices in clinical and interprofessional contexts. This is particularly relevant given the growing concerns with documentation and patient safety [25–27]. Regarding this specific group of students, we have already conducted interviews based on the questionnaire on the one hand, and on their individual notes on the other. The results of this study can broaden the understanding of students' note-taking with concrete examples and patterns, which may support development of teaching and nursing programmes.

## Conclusion

Academic literacy was represented by two disciplines within the intended curricula: nursing and medicine. Furthermore, expressions of professional literacy practices were represented in terms of documentation and communication (talking about professional texts with teachers, colleagues, various groups of patients, and other professions in relation to interprofessional teamwork), mainly in courses related to nursing. Expressions of academic writing practices within the intended curricula were primarily seen in the introductory nursing courses and in the last year, e.g., in the degree project course writing the bachelor's thesis. Those courses were mainly language based (lectures, course literature, writing assignments) and included learning objectives concerning the philosophy of science, and methodology, but also reference management. Regarding literacy practices, the results show that a majority of the students take notes at lectures on a regular basis and before examinations. For note-taking, they foremost use pen and paper and then computers/tablets. Thus, students must cope with three distinct literacy practices, each with its own norms for reading and writing (what to read and write, how to read and write, as well as for what purposes), often without guidance. According to the students' answers to open-ended questions, their purposes of note-taking were to facilitate learning and memory. Taken together, this study highlights the necessity for clearer integration and explicit teaching of both literacy types to prepare students effectively for academic success and professional nursing practice, where the purpose of documentation (aspect of professional literacy practices) is to communicate both with other health care professions as well as patients and their relatives.

## Supporting information

**S1 File. The original questionnaire (in Swedish), distributed via the Research Electronic Data Capture (REDCap) web application, to nursing students in their second semester of the study programme in nursing during the autumn of 2023.**
(PDF)

**S2 File. The author translation of original questionnaire (in English), distributed via the Research Electronic Data Capture (REDCap) web application, to nursing students in their second semester of the study programme in nursing during the autumn of 2023.**
(PDF)

**S3 File. The data (e.g., the responses to the open-ended questions) that were used in the qualitative content analysis, that was applied to analyze and describe the results on literacy practices.** The responses are displayed in original (Swedish) on one sheet and with author translation (English) on a second sheet.
(XLSX)

## Author contributions

**Conceptualization:** Maria Christidis, Nikolaos Christidis, Viveca Lindberg.

**Formal analysis:** Maria Christidis, Nikolaos Christidis, Viveca Lindberg, Kristina Gottberg, Carina Georg.

**Investigation:** Maria Christidis, Nikolaos Christidis, Viveca Lindberg, Kristina Gottberg, Carina Georg.

**Methodology:** Maria Christidis, Nikolaos Christidis, Viveca Lindberg, Kristina Gottberg, Carina Georg.

**Writing – original draft:** Maria Christidis.

**Writing – review & editing:** Nikolaos Christidis, Viveca Lindberg, Kristina Gottberg, Carina Georg.

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
