## [Decision Letter · Decision Letter 0]

13 Aug 2025

Dear Dr. Christidis,

Thank you for submitting your manuscript to PLOS ONE. After careful consideration, we feel that it has merit but does not fully meet PLOS ONE’s publication criteria as it currently stands. Therefore, we invite you to submit a revised version of the manuscript that addresses the points raised during the review process.

**ACADEMIC EDITOR: **

**To further enhance the study, consider:**Consider clarifying the sample and analysis methods for flowAssessing students' literacy skills: Incorporate objective evaluations of students' literacy skills to offer a more thorough insight into their capabilities.Exploring the impact on patient care: Examine how the incorporation of academic and professional literacy practices in nursing education influences patient care and outcomes.Longitudinal design: Contemplate a longitudinal approach to investigate how students' literacy practices develop throughout their nursing education and into their professional lives.**Key Recommendations:**The language can be tightened for clarity and conciseness; some sentences are long and complexBreak up long sentences and clarify definitions early in the text.Standardise terminology and proofread for typos and formatting issues.Reformat tables and ensure all data is clearly presented.Streamline the discussion to focus on the most important findings and their implications for nursing education.Make recommendations for curriculum development more direct and actionable in the conclusion**Overall Comment:**The manuscript is a valuable contribution to the field of nursing education. Furthermore, it presents original and relevant research, conducted and reported to a high standard. With minor revisions for language, formatting, and a stronger focus on practical implications, it will make a valuable contribution to the literature on literacy practices in nursing education and is suitable for publication.**Some potential areas for future research include:**Evidence-Based Practice education: Analysing the effectiveness of evidence-based practice education programs in improving nursing students' literacy skills and competencies.Customisable Education Programs: Creating tailored educational programs that address diverse learning needs and encourage the integration of academic and professional literacy practices.Interprofessional Collaboration: Exploring the role of literacy practices in promoting interprofessional collaboration.

We look forward to receiving your revised manuscript.

Kind regards,

Olugbenga Ige

Academic Editor

PLOS ONE

Journal Requirements: 

2. Please ensure that you have specified a) Did participants provide their written or verbal informed consent to participate in this study?

3. In consent please state in Ethics Method section and manuscript if it is written or verbal. If consent was verbal, please explain a) why written consent was not obtained, b) how you documented participant consent, and c) whether the ethics committees/IRB approved this consent procedure.

5. We note that your Data Availability Statement is currently as follows: [All relevant data are within the manuscript and its Supporting Information files and can be obtained from the authors]

Additional Editor Comments:

I have reviewed the reports submitted by the reviewers appointed to evaluate the research manuscript, and I am strongly convinced that the appropriate option is to request the author to make major revisions to the research manuscript. The report submitted by Reviewer 2 was comprehensive enough to confirm my decision. The author should address all the comments made in the report submitted, or provide a rebuttal where such revisions are no longer possible, to enable a re-evaluation of the research manuscript.

Reviewers' comments:

Reviewer's Responses to Questions

**Comments to the Author**

1. Is the manuscript technically sound, and do the data support the conclusions?

Reviewer #1: Yes

Reviewer #2: Partly

2. Has the statistical analysis been performed appropriately and rigorously?

Reviewer #1: Yes

Reviewer #2: Yes

3. Have the authors made all data underlying the findings in their manuscript fully available?

Reviewer #1: Yes

Reviewer #2: Yes

4. Is the manuscript presented in an intelligible fashion and written in standard English?

Reviewer #1: Yes

Reviewer #2: Yes

Reviewer #1: The paper was academically sound. The conclusion was drawn from the data provided from the results. The statistical analysis in table 2 was performed correctly. The manuscript was written in standard English.

Reviewer #2: Your manuscript explores the integration of academic and professional literacy practices in a Swedish undergraduate nursing program, combining curriculum analysis with a student survey on note-taking. The objectives of the study are clearly articulated, address a relevant gap, and employ an appropriate methodology. The language utilised is straightforward, though with minor errors, making the study accessible and comprehensible.

Nevertheless, the results fail to specify the extent, in numerical terms, to which participants demonstrate their ability to apply academic literacy, which is essential for evaluating the study's success. This data would undoubtedly provide the reader with a clearer understanding of the success rate of the literacy test conducted. To better grasp the findings, it would be beneficial to have precise definitions of academic and professional literacy.

Potential bias in student responses is noted: The response rate for the digital questionnaire was 40%, which may introduce bias if the characteristics or literacy practices of the responding students differ from those of non-responding students.

To further enhance the study, consider:

Consider clarifying the sample and analysis methods for flow

Assessing students' literacy skills: Incorporate objective evaluations of students' literacy skills to offer a more thorough insight into their capabilities.

Exploring the impact on patient care: Examine how the incorporation of academic and professional literacy practices in nursing education influences patient care and outcomes.

Longitudinal design: Contemplate a longitudinal approach to investigate how students' literacy practices develop throughout their nursing education and into their professional lives.

Key Recommendations:

The language can be tightened for clarity and conciseness; some sentences are long and complex

Break up long sentences and clarify definitions early in the text.

Standardise terminology and proofread for typos and formatting issues.

Reformat tables and ensure all data is clearly presented.

Streamline the discussion to focus on the most important findings and their implications for nursing education.

Make recommendations for curriculum development more direct and actionable in the conclusion

Overall Comment:

The manuscript is a valuable contribution to the field of nursing education. Furthermore, it presents original and relevant research, conducted and reported to a high standard. With minor revisions for language, formatting, and a stronger focus on practical implications, it will make a valuable contribution to the literature on literacy practices in nursing education and is suitable for publication.

Some potential areas for future research include:

Evidence-Based Practice education: Analysing the effectiveness of evidence-based practice education programs in improving nursing students' literacy skills and competencies.

Customizable Education Programs: Creating tailored educational programs that address diverse learning needs and encourage the integration of academic and professional literacy practices.

Interprofessional Collaboration: Exploring the role of literacy practices in promoting interprofessional collaboration.

**Do you want your identity to be public for this peer review?** For information about this choice, including consent withdrawal, please see our Privacy Policy

Reviewer #1: No

Reviewer #2: **Yes: ** Florence Fezeka Mafisa

---

## [Author Response · Author response to Decision Letter 1]

15 Sep 2025

Response to reviewers:

Intended and experienced literacy practices in a Swedish undergraduate nursing education.

Thank you for considering this manuscript for publication. We would like to thank the reviewers and you for the comments and suggested changes in order to improve the quality of this manuscript. The changes are highlighted with track changes. We hope that this reply and changes in the manuscript are sufficient and will answer your queries.

Maria Christidis 2025-08-24

Reviewer questions and replies:

Reviewer 1:

The paper was academically sound. The conclusion was drawn from the data provided from the results. The statistical analysis in table 2 was performed correctly. The manuscript was written in standard English.

Reply: Thank you for this feedback.

Reviewer 2 and academic editor:

Your manuscript explores the integration of academic and professional literacy practices in a Swedish undergraduate nursing program, combining curriculum analysis with a student survey on note-taking. The objectives of the study are clearly articulated, address a relevant gap, and employ an appropriate methodology. The language utilised is straightforward, though with minor errors, making the study accessible and comprehensible.

Reply: Thank you for this feedback. The language has been adjusted where necessary.

Nevertheless, the results fail to specify the extent, in numerical terms, to which participants demonstrate their ability to apply academic literacy, which is essential for evaluating the study's success. This data would undoubtedly provide the reader with a clearer understanding of the success rate of the literacy test conducted.

Reply: Thank you for your comment. This must be a misunderstanding. The aim of this study was the following: “Thus, the aim of this study was to analyze one of the Swedish undergraduate nursing programmes, with respect to possible content of relevance for academic and professional literacy. Secondarily, to describe note-taking as an aspect of literacy practices, from the perspective of nursing students” (see line 151-154). Also, “A questionnaire was developed to gather information on students' note-taking as an aspect of literacy practices during their education” (see line 294-295). In order to clarify this we changed the wording from “gather information on…” to “map students’…”.

To better grasp the findings, it would be beneficial to have precise definitions of academic and professional literacy.

Reply: Thank you. Academic literacy has been described on line 64-67 and we have made a small clarifying addition in the end of the sentence: “Here. academic literacy refers to differences in what people read and write, how they do it and for what purposes (why). These practices vary across social contexts, including disciplines and related programmes (1-3).” However, we have added to the description of professional literacy on line 107-11: Professional literacy involves questions such as: what kinds of texts are typical and recurring within a specific profession? What are characteristic features (content and form) of these texts? What are the purposes of these texts? The answers to these questions differ between professions (on engineers’ professional writing (14)).”

Potential bias in student responses is noted: The response rate for the digital questionnaire was 40%, which may introduce bias if the characteristics or literacy practices of the responding students differ from those of non-responding students.

Reply: Thank you. This has been addressed in the following lines: See line 570-571: “The response rate of the questionnaire was relatively small (40%) despite reminders. The small sample (n<100) allows us only to regard the quantitative result as indications”. Also, on line 580: “Thus, the results must be interpreted with caution”.

To further enhance the study, consider:

Consider clarifying the sample and analysis methods for flow

Reply: Thank you. This has been clarified in the analytical methods section we have written a complementary clarification of descriptive statistics. However, since the data was a survey for mapping students literacy practices, not a test (clarified as previously described), other desciptives than frequency distributions are not relevant – see lines 312, 345-346 “For closed questions frequency distribution was used, complemented with mean and range for age (38), while for the open-ended questions, a qualitative content analysis (39) was applied to analyze and describe the results on literacy practices.“

Assessing students' literacy skills: Incorporate objective evaluations of students' literacy skills to offer a more thorough insight into their capabilities.

Reply: Thank you. This is a misunderstanding, see previous comment: The aim of this study was the following: “Thus, the aim of this study was to analyze one of the Swedish undergraduate nursing programmes, with respect to possible content of relevance for academic and professional literacy. Secondarily, to describe note-taking as an aspect of literacy practices, from the perspective of nursing students” (see line 152-155). Also, “A questionnaire was developed to map students' note-taking as an aspect of literacy practices during their education” (see line 295-296). In order to clarify this we changed the wording from “gather information on…” to “map students’…”.

Exploring the impact on patient care: Examine how the incorporation of academic and professional literacy practices in nursing education influences patient care and outcomes.

Reply: Thank you. We have indications and can strengthen these indications by complementing the introduction concerning previous research (see line 124-137)

Longitudinal design: Contemplate a longitudinal approach to investigate how students' literacy practices develop throughout their nursing education and into their professional lives.

Reply: Thank you. This is the first part of a longitudinal project. Subsequently literacy in the nursing programme will be further explored in later semesters. This is in line with our previous project concerning literacy in the dental programme, where various aspects of literacy have been explored in terms of students’ clinical notes and students’ notes from lectures, as well as interviews with students

• Lindberg, V., Jounger, S. L., Christidis, M., & Christidis, N. (2020). Characteristics of dental note taking: a material based themed analysis of Swedish dental students. BMC Medical Education, 20(1), 511.

• Lindberg, V., Jounger, S. L., Christidis, M., & Christidis, N. (2021). Literacy as part of professional knowing in a Swedish dental education. BMC Medical Education, 21(1), 373.

• Christidis, N., Lindberg, V., Jounger, S. L., & Christidis, M. (2022). Early steps towards professional clinical note-taking in a Swedish study programme in dentistry. BMC Medical Education, 22(1), 676.

• Christidis, N., Lindberg, V., Helo, H., Koj, S., & Christidis, M. (2023). Swedish dental students’ clinical notes and reflections as part of a case-based examination–challenges for undergraduate education. Medical Research Archives, 11(10).

• Christidis, N., Tomasson, J., Rataghi, A., & Christidis, M. (2024). Preparation of dental and nursing professionals within Swedish higher education: navigating to confidence in literacies and professional knowledge. BMC Medical Education, 24(1), 1426.

Key Recommendations:

The language can be tightened for clarity and conciseness; some sentences are long and complex

Break up long sentences and clarify definitions early in the text.

Standardise terminology and proofread for typos and formatting issues.

Reply: Thank you, this has been adjusted accordingly.

Reformat tables and ensure all data is clearly presented.

Reply: Thank you. The tables have been adjusted to more clearly present the results.

Streamline the discussion to focus on the most important findings and their implications for nursing education.

Reply: Thank you. The discussion has been adjusted and streamlined

Make recommendations for curriculum development more direct and actionable in the conclusion

Reply: Thank you. This has been added in line 618-620, and in line 624-626 in order to highlight the need we see today (it is in bold text here to make it easier to identify) “Thus, students must cope with three distinct literacy practices, each with its own norms for reading and writing (what to read and write, how to read and write, as well as for what purposes), today often without guidance” together with the following part “Taken together, this study highlights the necessity for clearer integration and explicit teaching of both literacy types to prepare students effectively for academic success and professional nursing practice, where the purpose of documentation (aspect of professional literacy practices) is to communicate both with other health care professions as well as patients and their relatives.”

Overall Comment:

The manuscript is a valuable contribution to the field of nursing education. Furthermore, it presents original and relevant research, conducted and reported to a high standard. With minor revisions for language, formatting, and a stronger focus on practical implications, it will make a valuable contribution to the literature on literacy practices in nursing education and is suitable for publication.

Reply: Thank you for this feedback.

Some potential areas for future research include:

Evidence-Based Practice education: Analysing the effectiveness of evidence-based practice education programs in improving nursing students' literacy skills and competencies.

Customizable Education Programs: Creating tailored educational programs that address diverse learning needs and encourage the integration of academic and professional literacy practices.

Interprofessional Collaboration: Exploring the role of literacy practices in promoting interprofessional collaboration.

Reply: Thank you for this feedback. We completely agree, and that is our plan, therefore we have added a part on interprofessional collaboration and growing concerns with documentation and patient safety (lines 585-590) but also regarding the development of teaching and nursing programmes on line 606.

---

## [Editor Report · Decision Letter 1]

22 Sep 2025

Dear Dr. Christidis,

Thank you for submitting your manuscript to PLOS ONE. After careful consideration, we feel that it has merit but does not fully meet PLOS ONE’s publication criteria as it currently stands. Therefore, we invite you to submit a revised version of the manuscript that addresses the points raised during the review process.

We look forward to receiving your revised manuscript.

Kind regards,

Olugbenga Ige

Academic Editor

PLOS ONE

Journal Requirements:

Additional Editor Comments:

The authors should highlight all changes to the research manuscript in green. The amendments in "Response to Reviewers" are hard to identify without highlighting these changes.

---

## [Author Response · Author response to Decision Letter 2]

24 Sep 2025

The changes in the manuscript are now highlighted i green and the "Response to reviewers" has been updated with new line-numbers since they were changed when I removed the "track-changes function from MS Word".

I hope it is more clear now.

Best Maria

---

## [Editor Report · Decision Letter 2]

7 Oct 2025

Intended and experienced literacy practices in a Swedish undergraduate nursing education

PONE-D-25-32643R2

Dear Dr. Christidis,

We’re pleased to inform you that your manuscript has been judged scientifically suitable for publication and will be formally accepted for publication once it meets all outstanding technical requirements.

Kind regards,

Olugbenga Ige, PhD

Academic Editor

PLOS ONE
---

## [Editor Report · Acceptance letter]

PONE-D-25-32643R2

PLOS ONE

Dear Dr. Christidis,

I'm pleased to inform you that your manuscript has been deemed suitable for publication in PLOS ONE. Congratulations! Your manuscript is now being handed over to our production team.

Kind regards,

on behalf of

Dr. Olugbenga Ige

Academic Editor

PLOS ONE